# Chemical Structure–Antioxidant Activity Relationship of Water–Based Enzymatic Polymerized Rutin and Its Wound Healing Potential

**DOI:** 10.3390/polym11101566

**Published:** 2019-09-26

**Authors:** Tanja Pivec, Rupert Kargl, Uroš Maver, Matej Bračič, Thomas Elschner, Ema Žagar, Lidija Gradišnik, Karin Stana Kleinschek

**Affiliations:** 1Laboratory for Characterization and Processing of Polymers (LCPP), Faculty of Mechanical Engineering, University of Maribor, Smetanova 17, 2000 Maribor, Slovenia; tanja.pivec@um.si (T.P.); matej.bracic@um.si (M.B.); thomaselschner@gmx.de (T.E.); 2Institute of Paper, Pulp and Fibre Technology (IPZ) Graz University of Technology, Inffeldgasse 23, 8010 Graz, Austria; rupert.kargl@um.si; 3Institute for Biomedical Sciences, Faculty of Medicine, University of Maribor, Taborska 8, 2000 Maribor, Slovenia; uros.maver@um.si (U.M.); lidija.gradisnik@um.si (L.G.); 4Department of Pharmacology, Faculty of Medicine, University of Maribor, Taborska 8, 2000 Maribor, Slovenia; 5Department of Polymer Chemistry and Technology, National Institute of Chemistry Slovenia, Hajdrihova 19, 1000 Ljubljana, Slovenia; ema.zagar@ki.si; 6Graz University of Technology, Stremayrgasse 9, 8010 Graz, Austria

**Keywords:** rutin, polyrutin, aqueous enzymatic polymerization, chemical structure, antioxidant activity

## Abstract

The flavonoid rutin (RU) is a known antioxidant substance of plant origin. Its potential application in pharmaceutical and cosmetic fields is, however, limited, due to its low water solubility. This limitation can be overcome by polymerization of the phenolic RU into polyrutin (PR). In this work, an enzymatic polymerization of RU was performed in water, without the addition of organic solvents. Further, the chemical structure of PR was investigated using ^1^H NMR, and FTIR spectroscopy. Size-exclusion chromatography (SEC) was used to determine the molecular weight of PR, while its acid/base character was studied by potentiometric charge titrations. Additionally, this work investigated the antioxidant and free radical scavenging potential of PR with respect to its chemical structure, based on its ability to (i) scavenge non biological stable free radicals (ABTS), (ii) scavenge biologically important oxidants, such as O_2_•, NO•, and OH•, and (iii) chelate Fe^2+^. The influence of PR on fibroblast and HaCaT cell viability was evaluated to confirm the applicability of water soluble PR for wound healing application.

## 1. Introduction

Flavonoids are the largest group of polyphenolic compounds of plant origin, with a variety of biological effects in numerous mammalian cell systems, in vitro and in vivo. They have been shown to exert antioxidant, anti-inflammatory, anti-mutagenic, anti-HIV (human immunodeficiency virus), and wound healing promotional properties [1,2]. Rutin (RU) is one of the major and better known flavonoids [3]. It is best known for its antioxidant and anti-inflammatory properties; therefore, the recent studies of RU have been focusing mostly on its applications in cosmetic and pharmaceutical products [4,5]. RU is already commercially available as an orally administered supplement to support wound healing [3]. A recent in vivo study on rat dorsal wounds demonstrated that an RU-conjugated hydrogel enhanced wound healing significantly [3,6], which makes RU an interesting agent for topical wound treatment. However, a significant limitation in the use of RU that prevents its widespread use lies in its low bioavailability after oral administration, due to its very low water solubility [7] and poor absorption from the gastrointestinal tract [3]. To overcome this, we focused on a chemical modification of RU to increase its water solubility. It is known that the water solubility of flavonoids can be enhanced by their enzymatic polymerization [7]. Since such a reaction could have a significant impact on the antioxidant activity and pharmacological efficacy of flavonoids, the latter has to be evaluated [7,8]. The enzymatic polymerization of RU leads to the formation of polyrutin (PR) with variable properties related to the reaction conditions and formed polymeric structure. In recent years, laccase–catalyzed polymerizations gained attention due to the environmentally benign process of polymer synthesis in water without the use of hydrogen peroxide, which is the usual oxidizing agent for the polymerization of polyphenols with peroxidases [8]. Due to the poor solubility of RU in water, previous studies dealt with the polymerization in aqueous organic solvent mixtures [7,8,9]. With the prospect of using PR for wound healing, as well as for other topical applications on the skin, the avoidance of organic solvents is, however, desirable. In this study, the aqueous synthesis of PR using laccase as a catalyst represents an environmentally more favorable process. RU acts as an antioxidant in various ways—with a direct scavenging of free radicals, and with the prevention of radical generation. The latter is enabled through different mechanisms, among them an iron-chelating activity [10]. It has been shown that natural chelators that reduce oxidative stress are preferred over synthetic ones [11]. While it has already been confirmed that RU forms complexes with transition metals [10], such experiments are yet to be performed for PR.

This work therefore aims at producing and characterizing aqueous, organic solvent free, enzymatically polymerized RU by the aforementioned parameters. To the best of our knowledge, such polymerization of RU, together with a systematic evaluation of its physico-chemical properties, has not been reported so far. To elucidate the structural and physicochemical properties, ^1^H NMR, Attenuated Total Reflectance-Fourier transform infrared spectroscopy (ATR-FTIR), size exclusion chromatography (SEC), and charge titration were performed with RU and PR. To confirm that the RU retains the most important properties needed for wound healing application after polymerization, the ABTS (2,2′-azino-bis(3-ethylbenzothiazoline-6-sulphonic acid), O_2_•, NO•, and OH• scavenging assays, Fe^2+^ chelation experiments and cell viability test using human skin derived cells were performed with RU and PR.

## 2. Materials and Methods

### 2.1. Synthesis of PR

PR was synthesized in ultra pure water (18.2 MΩ cm at 25 °C) from an ELGA PureLab water purification system (Veolia Water Technologies, Lane End, UK) by a modified method of Jeon et al. [8,12]. Briefly, 0.5 g of RU hydrate (Sigma-Aldrich, Schnelldorf, Germany) was dispersed in ultra pure water (50 mL) and enzymatic polymerization was initiated by the addition of 500 mg laccase from *Trametes versicolor* (0.5 U/mg, Sigma-Aldrich, Schnelldorf, Germany, molecular weight between 60 and 110 kDa, BRENDA:EC1.10.3.2) at room temperature for 24 h under gentle stirring and protection from light. The enzyme was precipitated by the addition of ammonium sulphate (750 g/L). The sample was centrifuged at 10,000 rpm and 4 °C for 25 min and the supernatant was recovered. The PR solution was dialyzed (membrane molecular weight cut-off 3500 Da) against ultra pure water for 24 h, and the water was exchanged four times. The remaining solution was lyophilized to give the solid polymer. To obtain hydrolyzed PR (polyquercetin) for ^1^H NMR measurements, 500 mg of PR were dissolved in 18 wt. % HCl in water and stirred for 24 h at room temperature. The precipitate was centrifuged at 10,000 rpm at room temperature for 10 min. The sediment was washed five times with ultra pure water and lyophilized to give the solid hydrolyzed PR insoluble in water.

### 2.2. Solubility Determination 

1000 mg/mL of PR was dispersed in solvent (ultra pure water, DMSO, DMA, DMF, pyridine, methanol, acetone, THF, and toluene) at room temperature for 24 h. Afterwards the sample was centrifuged at 10,000 rpm and room temperature for 25 min. The concentration of PR in the supernatant was determined by UV-Vis spectrophotometer (Cary 60 UV-Visible Spectrophotometer, Agilent, Santa Clara, CA, USA) by quantification of the absorption band at 260 nm. Determination of the concentration in the supernatant was performed in three parallels. 

### 2.3. ^1^H NMR Spectroscopy

^1^H NMR analysis of RU, PR, and hydrolyzed PR were performed on a Bruker BioSpin GmbH (Rheinstetten, Germany), at 296 K with the number of scans 16 after dissolution in DMSO-d_6_ at a concentration of 125 mg/mL. 

### 2.4. ATR-FTIR Spectrometry

The ATR-FTIR spectra of samples were recorded using a Perkin Elmer Spectrum Two IR spectrometer (Waltham, Massachusetts, USA). The spectra were collected in the range of 500–4000 cm^−1^ at room temperature. 16 scans were performed for each measurement with a resolution of 0.5 cm^−1^.

### 2.5. Size-Exclusion Chromatography (SEC)

SEC measurements were performed at room temperature on a Hewlett-Packard modular system consisting of a pump and a degasser series 1100, coupled to a UV detector, operating at 280 nm (Agilent Technologies 1260 Infinity, Santa Clara, CA, USA). Separation of samples was carried out on a PolarGel-L 8 μm column with a pre-column (300 mm length and 7.5 mm i.d., molar mass range: Up to 30 kDa, Agilent Technologies, Santa Clara, CA, USA) in 0.05 M LiBr in *N*,*N*-dimethylacetamide at an eluent flow rate of 1 mL/min. The mass of the sample injected onto the column was typically 1 × 10^−4^ g, whereas the solution concentration was 1 × 10^−3^ g/mL. The SEC column was calibrated using polystyrene standards. PR and RU molecular weight averages were calculated based on calibration of the SEC column with polystyrene standards of defined molecular weight and narrow molecular weight distribution.

### 2.6. Potentiometric Charge Titration

The pH-dependent potentiometric titrations were performed for RU and PR by dissolving a 100 mg sample in 30 mL of ultra pure water with a very low carbonate ion content (c = 10^−6^ mol/L). This was achieved by boiling and subsequent cooling of the water in a nitrogen gas atmosphere. The solution was titrated from pH = 2 to pH = 11 using 0.1 mol/L hydrochloric acid (Fluka Analytical, Seelze, Germany) and 0.1 mol/L potassium hydroxide (J.T Baker, Dilut-It, Fair Lawn, New Jersey, USA). The ionic strength of the solution was set to 0.1 mol/L using potassium chloride. The titrants were added to the system in a dynamic mode using a double burette Mettler Toledo T70 automatic titration unit. The pH value was measured using a Mettler Toledo InLab Routine (L) combined glass electrode. Details on the measurements and the calculation of the charges can be found elsewhere [13].

### 2.7. Antioxidant Activity

The antioxidant activity of RU and PR was determined using different in vitro methods. 

#### 2.7.1. ABTS Radical Scavenging Assay

ABTS, 2,2′-azinobis(3-ethylbenzothiazoline-6-sulfonic acid) diammonium salt (Sigma-Aldrich, Slovenia) was used as a free radical. ABTS and potassium persulfate (Sigma-Aldrich, Slovenia) were dissolved in ultra pure water at a final concentration of 7 and 2.75 mM, respectively. The solution was allowed to stand in the refrigerator for at least 12 h before use in order to produce ABTS radicals (ABTS•). The ABTS• solution was diluted with phosphate buffered saline (PBS) buffer to an absorbance of 0.7 ± 0.0025 at 734 nm. RU solution in methanol (0.1 mL) or PR solution in ultra pure water (0.1 mL) was added to 3.9 mL of ABTS• solution in PBS buffer and the decrease in absorbance was determined at 0 min and after 15 min. Percentage ABTS• scavenging activity was calculated according to Equation (1):% ABTS• scavenging activity = ((Ac − As)/Ac) × 100,(1)
where Ac is the absorbance of the control and As is the absorbance of the sample. The half maximal inhibitory concentration (IC_50_) values represent the amount of tested substance where 50% of the radicals were scavenged by the test samples. All the experiments were conducted in duplicate. Data are presented as mean ± standard deviation.

#### 2.7.2. Superoxide Radical (O_2_•) Scavenging Assay

The O_2_• scavenging assay was done according to a previously reported method [14]. The non-enzymatic phenazine methosulfate-nicotinamide adenine dinucleotide (PMS/NADH) system generates O_2_• that reduces nitroblue tetrazolium (NBT) into a purple-colored formazan. The reaction mixture contained 1 mL of 156 µM NBT (Sigma-Aldrich) dissolved in a phosphate buffer (pH 7.4), 1 mL of 486 µM NADH (Sigma-Aldrich) dissolved in a phosphate buffer (pH 7.4), 0.1 mL of sample (RU in aceton, PR in ultra pure water) solution (0–1000 μg/mL) and 0.1 mL of 60 µM PMS (phenazine methosulfate, Sigma-Aldrich) solution. After the incubation of the reaction mixture for 5 min at 25 °C, the absorbance was taken at 560 nm. Ascorbic acid (AA), dissolved in ultra pure water in concentrations corresponding to that of the samples, was used as a reference compound. Percentage of O_2_• scavenging activity was calculated according to Equation (1).

#### 2.7.3. Nitric Oxide (NO•) Scavenging Assay

At physiological pH, NO• generated from aqueous sodium nitroprusside (SNP) solution interacts with oxygen to produce nitrite ions, which were measured by the Griess Illosvoy reaction [14]. The reaction mixture contained 2 mL 10 mM SNP in a phosphate buffer (pH 7.4) and 0.5 mL of sample (RU in acetone, PR in ultra pure water) solution (0–1000 μg/mL) were incubated for 150 min at 28 °C. After the incubation period, 0.5 mL of Griess reagent (Sigma-Aldrich, Schnelldorf, Germany) was added to the reaction mixture. The absorbance of pink chromophore was read at 454 nm. AA dissolved in ultra pure water in concentrations corresponding to that of the samples was used as a reference compound. Percentage of NO• scavenging activity was calculated according to Equation (1).

#### 2.7.4. Hydroxyl Radical (OH•) Scavenging Assay

The method is based on the specific reaction of deoxyribose with OH• generated from ascorbic acid/Fe2+/EDTA (ethylenediaminetetraacetic acid). This produces malondialdehyde (MDA), which can be evaluated by reaction with thiobarbituric acid (TBA) to form a pink chromogen [15]. The scavenging capacity for OH• was measured according to the modified method of Halliwell et al. [16,17]. Stock solutions of EDTA (1 mM), FeCl_3_ (10 mM), ascorbic acid (1 mM), H_2_O_2_ (10 mM), and 2-deoxy-D-ribose (10 mM) were prepared in ultra pure water. The assay was performed by adding in sequence 0.1 mL EDTA, 0.01 mL FeCl_3_, 0.1 mL H_2_O_2_, 0.36 mL deoxyribose, 1 mL of sample (RU in acetone, PR in ultra pure water) solution (100–1000 μg/ mL), 0.33 mL of phosphate buffer (pH 7.4), and 0.1 mL ascorbic acid. The mixture was incubated at 37 °C for 1 h. Then, 1 mL each of 10% TCA (trichloroacetic acid) and 0.5% TBA were added to the reaction mixture and kept at 100 °C for 20 min to develop the pink chromogen. Absorbance was measured at 532 nm. Ascorbic acid (AA), dissolved in ultra pure water in concentrations corresponding that of the samples, was used as a reference compound. Percentage of OH• scavenging activity was calculated according to Equation (1). 

#### 2.7.5. Fe^2+^ Chelating Activity

The chelating effect was determined according to the method described by Parwani et al. [18] with some modifications. Briefly, 3 mL of various RU concentrations (0.5–75 mg/mL) in methanol and PR in ultra pure water were taken. Then, 2 mM FeSO_4_·7H_2_O (0.05 mL) was added. The reaction was initiated by adding 0.2 mL of 5 mM ferrozine solution (3-(2-pyridyl)-5,6-diphenyl-1,2,4-triazine-4′,4″-disulfonic acid sodium salt, Sigma-Aldrich). Then, the mixture was shaken vigorously and left at room temperature for 10 min. Absorbance of the solution was measured spectrophotometrically at 562 nm. FeSO_4_·7H_2_O solutions were considered as a blank, whereas FeSO_4_·7H_2_O and ferrozine solution were used as a control. All determinations were carried out in duplicate. The inhibition percentage of the ferrozine—Fe^2+^ complex formation was calculated according to Equation (1).

### 2.8. Cell Viability

A cell viability test was performed using human skin-derived fibroblasts (ATCC-CCL-110, Detroit 551, LGC Standards, Manassas, VA, USA) and HaCaT cells (human-derived immortalized keratinocytes). For this purpose, solutions of RU and PR were prepared in phosphate buffered saline (PBS) in concentrations 5.4 and 0.54 μg/mL. The test was performed using P96 microtiter plates. To determine the highest possible safe concentration and to avoid sediments of RU due to its low water solubility, dilutions with the factor ten were used consecutively.

Quadruplicates of each dilution (10 and 100 in our case), as well as of the control sample (Advanced Dulbecco’s modified Eagle’s medium—ADMEM, Gibco, Grand Island, NY, USA, supplemented with 5 wt. % fetal bovine serum—FBS, Gibco, Grand Island, NY, USA), were used. Each well was filled with cell suspension containing 60,000 cells and, after 24 h, when the cells attached and formed a monolayer, it was supplemented with the dissolved RU hydrate and PR sample solutions of desired dilution. Cell viabilities were observed after 24 h of incubation at 37 °C and 5 wt. % CO_2_. Cell viability was determined via the reduction reaction of the tetrazolium salt MTT (3(4,5 dimethylthiazolyl-2)-2,5-diphenyltetrazolium bromide), as determined by measuring the absorbance at 570 nm [19,20].

### 2.9. Statistical Analysis

Data presented in the present study are means ± standard deviation of the mean. The statistical analysis for the MTT assay was performed using the one-way analysis of variance (ANOVA) using Excel 2013. A *p* < 0.05 was considered as significant.

## 3. Results and Discussion

### 3.1. Chemical Structure of Polyrutin

The present study investigated the simple, low cost, and green polymerization process only in water in the homogeneous phase, without any precipitation or undissolved residue of RU after 24 h. The precipitation of laccase with ammonium sulphate, which was used in the cleaning phase, is a well-known procedure for the isolation of enzymes from fungi [12,13].The concentration of enzyme (activity) and the time of reaction were chosen to be in the ranges examined in other studies [8,19,21,22], namely 5000 U/L and 24 h. However, Muniz-Mouro et al. examined the effect of the concentration of enzyme from *Trametes versicolor* on the antioxidant activity of the PR synthesized in water–ethanol mixture, and found that the PR obtained in the reaction with low laccase activity (1000 U/L) exhibited higher inhibitory activity of xanthine oxidase compare to PR obtained in the reaction with high laccase activity (10,000 U/L), which they attributed to the different chemical structure of PRs obtained at different concentrations of the enzyme [22]. In this study we examined only one intermediate concentration (5000 U/L), but in the future it would be interesting to examine the effect of the concentration of the enzyme, as well other reaction parameters (source of enzyme, time of the reaction, the mass ratio between RU and enzyme, pH, and temperature) on the chemical structure and antioxidant properties of PR.

The optimum pH value for laccases varies depending on the substrates employed. The acidic pH region is the optimum for most fungal laccases for the phenol substrates [23,24]. However, the pH value of the reaction in our study was not set to a specific value, but it was monitored during the reaction. Specifically, the pH of the dispersion of RU in ultra pure water before the addition of the enzyme was 7.19. After the addition of the enzyme, the pH value decreased to 5.85 after 1 min of stirring, to 4.86 after 5 min and to 4.24 after 2 h. Since the oxidation of phenol substrates involves a proton transfer from the substrate, the drop in pH can indicate the immediate onset of the oxidation reaction. In the next 2 h the reaction media reached a pH value of 3.87 and it remained constant for the next 6 h. Then the pH value started to increase slowly to 7.12 in the next 8 h, indicating the completion of the reaction. The pH remained constant for the next 8 h. A pH adjustment, for example to a value above the pKa (8.1, Figure 4a), where RU shows higher solubility, would most likely have a negative effect on the reaction since laccase from *Trametes versicolor* is inactive in this pH range [25]. The RU polymerization in alkaline-pH could be successful with laccases from other sources (bacteria, plant, and fungi), which have an optimal pH activity from 7–10 (see BRENDA database, EC1.10.3.2), as already confirmed by Uzan et al. who polymerized RU with the *Myceliophthora thermophila* laccase (commercially available as Novozymes Suberase^®^) at pH 7.5 [26].

#### 3.1.1. Solubility of PR

The yield of the reaction was 57 wt. %, (0.28 g of PR was obtained from 0.5 g of RU hydrate). Although the RU monomer showed very low water solubility (0.125 mg/mL) [26], the resulting polymer was well soluble in water. The solubility of PR in water at room temperature was determined photometrically to be 753 ± 27 mg/mL. Very high solubility in water (600 mg/mL) of PR compared to RU was also observed by Anthoni et al., but their synthesis of PR was performed in aqueous organic media. They explained the differences in solubility by evaluating the hydrogen bonds in a molecular modeling study. They concluded that RU has a folded structure where L-rhamnose and the ring A came closer together, whereas PR has an unfolded structure in which sugars offer a large contact with the surrounding solvent (water). For this reason, the number of intermolecular hydrogen bonds between the PR and water is higher than between RU and water, explaining the higher solubility of PR in water [9]. Additionally, PR solubility in solvents with different polarity index [27] was determined photometrically, and the results were combined with the results obtained by Kurisawa et al. (Table 1), who evaluated the solubility of PR only by visual observation. Kurisawa et al., who synthesized PR in an aqueous organic media, found that PR is soluble in water, DMF, DMSO, methanol, and, partially, in pyridine [8]. Contrary to literature, our sample of PR is well soluble only in water and DMSO. The solubility of PR in solvents with high polarity index is the consequence of its chemical structure. With the examination of the chemical structure of PR, we demonstrated that the main chain of PR consisted of non-polar phenolic rings surrounded by sugar units (rutinose; results of ^1^H NMR spectroscopy below). Due to the exposure of polar hydroxyl groups on the sugar units, PR is soluble in solvents with a high polarity index. In comparison with the PR synthesized in an aqueous organic media, PR synthesized in water is soluble only in solvents with higher polarity index, which means that the reaction media has an impact on the number of exposed polar hydroxyl groups, which are the result of different linkages between monomer units.

#### 3.1.2. ^1^H NMR Spectroscopy

^1^H NMR spectroscopy was applied to reveal the molecular structure of RU and PR synthesized in water, and to compare those results with the current literature data about the chemical structure of PR synthesized in aqueous organic solvent mixtures. In the ^1^H NMR spectrum of RU (Figure 1, top), all resonances of aromatic CH moieties (6′, 2′, 5′, 8, and 6) were visible from 6 to 8 ppm. Moreover, the aliphatic CH groups of the sugar units could be observed. In particular, positions 1″, 1‴, and 6‴ were well-separated signals appearing at 5.02, 4.56, and 1.15 ppm. The residual CH signals of rutinose were detected between 3 and 4 ppm. All resonances of aromatic hydroxylic groups (OH-5, OH-7, OH-3′, and OH-4′) of RU were also visible in the spectrum from 9 to 13 ppm. The aliphatic hydroxylic groups of the sugar units (OH-2″, OH-3″, OH-4″, OH-2‴, OH-3‴, and OH-4‴) could be observed from 4 to 5.5 ppm. The assignment of the resonances was performed according to the literature [19]. Up to now, the complete structure of PR could not be clarified unambiguously. However, the signals of the rutinose moiety were observed (Figure 1, centre). The resonances of the aromatic CH groups were broad, and possessed very low intensity, due to a fast relaxation mechanism caused by low flexibility. On the one hand, the hydrophobic polyquercetin (PQ) main chain was less solvated by the polar solvent DMSO-d6. On the other, this backbone was stiff compared to flexible rutinose side chains. Thus, relaxation of the aromatic part of the molecule was comparable to those in the solid state, while the hydrophilic sugar moieties caused sharp signals of a typical NMR solution. 

Nevertheless, the signals of the aromatic polymer backbone could be detected after acidic hydrolysis, which caused the cleavage of the rutinose side chains. Subsequently, a ^1^H NMR spectrum of the obtained PQ was recorded (Figure 1, bottom). Due to the lower molecular weight and the increased mobility of the main chain, the CH and OH resonances between 6 and 13 ppm became visible. Although the structure of PR could not be clarified completely, we can observe some signals that indicate which atoms could be involved in the polymerization of RU. In the spectrum of PQ the resonances of aromatic OH-5 and OH-7 were visible at 12.40 ppm and 10.75 ppm, indicating free meta-hydroxyl groups in the A-ring. The resonances of aromatic CH moieties (8 and 6) were also visible at 6.92 ppm and 6.21 ppm. Positions OH-3′ and OH-4′ appearing at 9.50 were not well separated signals, and were not very intense, indicating involvement of at least one –OH group in the polymerization of RU. The resonance of aromatic CH-5′ was visible at 7.72 ppm. The positions CH-6′ and CH-2′ were not very intensive in the region from 8.00 to 7.50 ppm, indicating the involvement of both positions in the polymerization. 

Although the chemical structure of PR could not be clarified completely, it could be concluded that the B-ring was undoubtedly more involved in the polymerization than the A-ring, wherein the positions CH-2′ and CH-6′, and at least one hydroxyl group (position OH-3′ or OH-4′) on the B-ring, were involved in the polymerization, and this could lead to the linking of RU monomer units with different sides and, consequently, to the formation of different chemical structures of PR. Furthermore, different molecular structures after the enzymatic polymerization of such a complex molecule as RU is, are certainly expected, as it was already shown by Muniz-Mouro et al. [22] and Anthoni et al. [9]. 

#### 3.1.3. ATR-FTIR Spectroscopy

To reveal the structure of RU, PR, and hydrolyzed PR additionally, FTIR spectra were taken and are presented in Figure 2. All spectra were normalized first and then shifted on the transmission axis to separate them for improving the transparency between the spectra. The IR analysis of the polymers (PR and PQ) shows a broadening of the absorption bands, reflecting that the polymer structure was more rigid compared to the monomer, due to the many new linkages between the monomer molecules that hinder the bond vibrations sterically. The intensities of the absorption bands between 1000 and 1650 cm^−1^ of the PR tended to weaken, which also suggests several new linkages that occurred during the polymerization (Figure 2, top). The increased intensity of the same absorption bands (between 1000 and 1650 cm^−1^) could be observed after acidic hydrolysis, which caused the cleavage of the rutinose side chains and, consequently, increase the vibration of bond in the main polyphenol chain (Figure 2, bottom). The bands between 3300 and 3500 cm^−1^ belong to phenolic and sugar OH vibrations. The peak at 1650 cm^−1^ was ascribed to the carbonyl vibration, and the peaks at 1600, 1550 and 1500 cm^−1^ to the C=C vibration of aromatic groups. The bands between 1000 and 1300 cm^−1^ were formed due to the various vibration modes, such as C–H, C–O, and C=O [8,12]. All samples show these peaks. In some other studies dealing with the polymerization of quercetin (an aglycon of RU) a new peak at 1720 cm^−1^ belonging to the C=O vibration has been reported [21,28]. This can be attributed to the intermediate step involving the formation of a quinone. This peak was not detected in our sample of PR or PQ. The intensity of the peak at 1050 cm^−1^ of the PR also increased, indicating the formation of new ether bonds (C–O–C) or carbon–carbon single bonds (C–C) between monomer units. Due to the fact that the laccase catalyzed hydrogen abstraction reactions from phenolic resulting in corresponding phenoxy radicals [29], and that the peak for the C=O vibration indicating the formation of the quinone was not observed, we could conclude that only, (or at least mostly), new ether bonds (C–O–C) were formed between monomer units. 

#### 3.1.4. SEC Analysis

Molecular weight characteristics of PR were studied using SEC/UV. Number–average molecular weight (Mn) for RU was 570 g/mol, weight–average molecular weight (Mw) was 590 g/mol, and dispersity (Mw/Mn) therefore was 1.03. Number–average molecular weight (Mn) for PR was 6600 g/mol, weight–average molecular weight (Mw) was 14,890 g/mol, and dispersity (Mw/Mn) was 2.2. The size exclusion chromatogram of PR (black curve in Figure 3) shows a unimodal molecular weight distribution with the peak apex a lower elution volume compared to RU (red curve in Figure 3). This result indicates that the polymerization of RU occurred and led to a polymer with higher molecular weight. According to the currently available research studies about the molecular weight averages of PR, only Kurisawa et al. [8], who synthesized PR in the mixture of methanol and a buffer solution, obtained somewhat higher molecular weight averages (15,800 g/mol) [8]. 

To the best of our knowledge, only two studies that investigate the chemical structure of PR more specifically have been published so far [9,30]. Uzan et al. dealt with the synthesis of PR in an aqueous mixture of glycerol/ethanol/buffer using laccases of different origins. The authors found that the structures of oligorutins (only dimers and trimers) were different, depending on the laccase origin. They also propose some preliminary chemical structures based on their findings. Unlike our findings, they propose the involvement of the A-ring in the polymerization, and the formation of quinone intermediates, as well as the formation of carbon–carbon single bonds (C–C) and ether bonds (C–O–C) between monomer units [30]. Anthoni et al. synthesized PR in methanol/water, and they found similar structural characteristics, except that the weight-average molecular weight (Mw) was lower (3,900 g/mol) [9]. From our investigation of the chemical structure of PR we can conclude that the polymerization of RU in water with the enzyme laccase from *Trametes versicolor* led to a polymer with a weight-average molecular weight of 14,890 g/mol, where the monomer units are coupled together mostly on the B-rings with the C–O–C bridges without the formation of quinones. 

#### 3.1.5. Potentiometric Titration

The pH dependent potentiometric titration isotherms of RU and PR are shown in Figure 4a. One can observe that the functional groups of RU protonate/deprotonate in the pH region of 6 ≤ pH ≤ 10, exhibiting one pKa value at half the neutralization point (pKa = 8.1). This is in good agreement with the data obtained by other authors regarding the pKa values of hydroxyl groups of RU [31]. In the case of PR, one can observe a much broader protonation/deprotonation range of 5 ≤ pH ≤ 11 and a pKa shift to 8.4. 

This can be attributed to the polymerization reaction. The RU molecule has phenol hydroxyl groups on the B-ring in position 3′ and 4′ and on the A-ring in position 5 and 7. The pKa of RU is, therefore, a combination of the pKa values of the B-ring and the A-ring phenolic hydroxyl groups. In the case of PR, several possible polymerization sites were located on the B-ring, which results in a vast number of possible PR structures. These new structures change the chemical environment of the phenol hydroxyl groups and, with that, their protonation and deprotonation character as well. Thus, a higher number of different pKa values is expected for PR, which results in a broader pH dependent protonation and deprotonation interval.

Besides the protonation/deprotonation changes, one can also observe changes in the total amount of phenol hydroxyl groups between RU and PR (Figure 4b). The RU molecule exhibited a total charge of 2.9 mmol/g, while the PR exhibited a total charge of 2.3 mmol/g. This charge could be directly attributed to the reduced amount of phenol groups, which we can observe from the correlation graph depicted in Figure 4c. The correlation between the reduction of free phenol hydroxyl groups after polymerization determined using ^1^H NMR, with the reduction of total charge after polymerization determined using potentiometric titrations for RU and PR indicate that the results correlated well. Namely, a 20% difference in total charge after polymerization was determined using potentiometric titrations, and this result is in agreement with the result of ^1^H NMR, where we assumed that one of four phenol hydroxyl group in the RU molecule, which was equal to 25% of the phenol hydroxyl group in RU molecule, was involved in the polymerization. This finding is important for further interpretation of the below described results, related to the wound healing performance.

### 3.2. Chemical Structure—Antioxidant Activity Relationship

Chronic wounds contain elevated levels of reactive oxygen and nitrogen species. The overproduction of free radicals, together with the iron ions’ accumulation, perpetuates the inflammatory phase, resulting in severe tissue damage. For this reason, the introduction of antioxidants appears to be a promising strategy to promote normal wound healing [18], and the various antioxidant agents, including flavonoids, have been shown to exert beneficial effects on the wound healing rate [32]. In the literature, some information is available on the antioxidant activity of RU and PR, which was synthesized in aqueous organic solvent mixtures. Interestingly, in some of the studies, the antioxidant activity of PR is significantly increased compared to the monomers [8,30]. The observation that polymerization enhances the antioxidant capacity is consistent with the fact that natural high-molecular-weight polyphenols of plants are much stronger antioxidants than low-molecular-weight polyphenols. The proximity of several aromatic rings and hydroxyl groups are more important for free radical scavenging than specific functional groups of natural phenols [8,33]. Contradictorily, some other studies found the antioxidant activity decreased with the polymerization of RU due to occupancy of the phenol hydroxyl groups [7,9]. This behavior could be due either to the used method of antioxidant activity determination, due to the degree of polymerization, or the type (C–C or C–O bridges) and position of these linkages [7].

Considering the differences between the chemical structure of PR synthesized in water, found in this work, compared to the chemical structures when synthesized in aqueous organic solvent mixtures [7,8,9], the determination of antioxidant activity is necessary. In this work, the antioxidant activity and free radical scavenging potential of RU and PR were tested using different in vitro methods, since the flavonoids are known to be able to prevent injury caused by free radicals in different ways. One way is a direct scavenging of the free radicals, where the flavonoids are oxidized by the radicals, resulting in more stable, less-reactive radicals, while the second way is the prevention of radical formation (by interfering with the enzymes, i.e., nitric-oxide synthase; by preventing leukocytes’ adhesion to the endothelial wall; by Fe^2+^ chelation).

The direct scavenging ability of free radicals is connected with the chemical structure of flavonoids and their redox potential. Due to their low standard redox potentials (E° < 1000 mV), flavonoids are able to reduce most oxidizing free radicals relevant to the biological systems thermodynamically [34]. The most important structural features, important for antioxidant activity of RU and other similar flavonoids, include the presence of 2,3 unsaturation in conjugation (shaded red in Figure 5) with a 4-oxo-function in the C-ring (shaded green in Figure 5), the presence of functional groups capable of binding transition metal ions (shaded blue) and a catechol group on the B ring (dihydroxylated B ring shaded yellow in Figure 5) [35]. The highest impact on antioxidant activity has the presence of a catechol group on the B ring, which is capable of donating hydrogen or electron readily to stabilize the radical species, due to the lowest redox potential of the catechol functional group in the RU molecule (233 mV, [36]). The redox potential of resorcinol moiety (ring A) in an RU molecule is 845 mV [36], meaning that the A ring is not a good electron donor, and, consequently, it has the minor contribution of ring A on the antioxidant activity of RU [37].

The first method used for the evaluation of direct scavenging of free radicals for RU and PR was an ABTS• scavenging assay. Although ABTS• are not present in the wound or human body, the ABTS method, among other free radical scavenging methods, is one of the most frequently used methods, because it is rapid, simple (i.e., not involved with many steps and reagents), and inexpensive [38]. The result of evaluation of the scavenging ability of ABTS• for RU and PR is shown in Figure 6a, where we can observe the decrease of antioxidant activity after RU polymerization. While the IC_50_ of RU was 1.1 ± 0.02 µg/mL, it was ten times higher for PR (IC_50_ = 10.2 ± 0.27 µg/mL). The reason for this could be the involvement of the B ring in the polymerization, and occupation of hydroxyl groups on the B ring after the polymerization, since the ABTS•, which has a redox potential 680 mV [39], could be reduced only by the catechol group on the B ring (E° = 233 mV). Interestingly, no scavenging ability of superoxide radicals with similar redox potential to ABTS• (E° = 650 mV [40]) could be observed for RU in Figure 6b. Even the pro-oxidant activity of RU was visible, which could be the consequence of the instability of RU phenoxyl radicals (the end product of radical scavenging by RU), which oxidize NADH rapidly in the in vitro testing system, resulting in extensive oxygen uptake and O_2_• formation [17]. Since the pro-oxidant activity is thought to be directly proportional to the total number of hydroxyl groups in a flavonoid molecule [17], the improvement of the O_2_• scavenging ability for PR could be connected with the lower number of free hydroxyl groups after polymerization, as well as with the higher stability of PR phenoxyl radicals. The latter could also be the reason for the almost equal NO• scavenging ability of PR (IC_50_ = 233 µg/mL) and RU (IC_50_ = 207 µg/mL; Figure 6c), meaning that the reducing of free hydroxyl groups after the RU polymerization did not reduce the NO• scavenging ability of RU, and it even improved in the higher concentration range from 600–1000 µg/mL. Regarding the IC_50_ value of positive control, AA (455 µg/mL), RU and PR had higher ability to scavenge the NO•, in spite of the lower redox potential of AA (212 mV [36]) compared to the catechol group on the B ring (E° = 233 mV). RU and PR also showed similar ability to scavenge OH• (Figure 6d), which was, among ROS, the most damaging radical, with a very high redox potential 2310 mV. The IC_50_ values of RU and PR were 1440 and 1330 μg/mL, respectively. The control substance AA showed the lowest IC_50_ value (238 μg/mL). Here also, the polymerization of RU and the occupation of free hydroxyl groups did not influence the inhibition of OH• generated with deoxyribose degradation in a Fe^3+^-EDTA-ascorbic acid and H_2_O_2_ reaction mixture. However, the scavenging of OH• in the used in vitro system could take place by two antioxidative mechanisms: One that suppresses the generation of OH•, and the other that scavenges them. In the former, the antioxidant may ligate to the metal ion, which otherwise reacts with H_2_O_2_ to give rise to OH•. Molecules that could inhibit deoxyribose degradation were those that could chelate irons and render them inactive or poorly active in a Fenton reaction, which will also be effective in wound healing [3]. Therefore, the total values of the OH• scavenging ability for RU and PR could be the summing of direct scavenging of OH• and the suppressing of the generation of OH• by the chelation. In the case of RU, the higher contribution to the total scavenging ability had the higher total number of free hydroxyl groups, but, in the case of PR, the higher contribution to the total scavenging ability had improved iron-chelating activity after polymerization. The latter was confirmed with the Fe^2+^-chelating ability test of RU and PR, presented in Figure 6e. At low concentrations (<5 mg/mL) RU showed better iron chelation than PR. However, at higher concentrations (>5 mg/mL) PR showed better iron chelation, that increased up to 95% at 10 mg/mL, while the iron chelation activity of RU increased up to 98% at five times higher concentration (50 mg/mL). This improvement of iron-chelating activity after polymerization was certainly correlated with the chemical structure of PR. The proposed binding sites for metals such as Fe^2+^ to RU, were the catechol moiety in ring B and the 4-oxo, 5-hydroxyl groups between the heterocyclic and the A rings (shaded blue in Figure 5). With the structural investigation of PR we found that the hydroxyl group on the position C5 (A-ring) was free, representing (together with the carbonyl group) a possible location for the binding of Fe^2+^. Although one of the hydroxyl groups in the B-ring was involved in the polymerization, the improvement of the Fe^2+^-chelating activity after polymerization could be the consequence of the formation of new possible intramolecular locations for the binding of Fe^2+^ as a result of the convergence of monomer units. 

Finally, we have to highlight that the ability of PR to scavenge all biologically important radicals (O_2_•, NO•, and OH•) and chelate Fe^2+^ in in vitro testing systems were retained or improved after polymerization. Moreover, it is, in contrast to RU, water soluble, making it more useful for applications as coatings or topological treatments as often required for wound treatment. 

### 3.3. Cell Viability 

Figure 7 shows the results from cell viability testing determined by the MTT assay.

Figure 7 shows the results obtained from the viability tests using fibroblasts from the dermis and HaCaT cells from the epidermis. According to literature [37], an appropriate concentration of RU to be applied on a wound in order to have a positive therapeutic effect on healing, should be approximately 50 µg/mL. Although, the water solubility of RU was 125 µg/mL, the present of salt in cell media decreased the solubility of RU and caused the sediments of RU particles of various sizes on the bottom of the microtitre plates, preventing any reliable measurements of cell viability at the concentrations higher than 5.4 µg/mL. For this reason, the MTT test was performed at concentrations 5.4 and 0.54 µg/mL for both substances.

The positive effect of RU and PR on the viability of fibroblasts compared to the control was proven at 5.4 µg/mL. Especially, the viability of the fibroblasts increased outstandingly in the presence of RU at a concentration of 5.4 µg/mL. The difference in the effect of RU and PR on the viability of fibroblasts could be related to their different chemical structure, as it has already been shown that different flavonoids with minor differences in the chemical structure can have a completely different influence on the cellular system (inhibitory or stimulatory) through direct action on various signaling pathways, such as phosphoinositide 3-kinase, Akt/protein kinase B, mitogen-activated protein kinase, tyrosine kinases, and protein kinase C [41]. However, at 0.54 µg/mL PR, but not RU, showed a significant positive influence on the viability of fibroblasts. For HaCaT cells, only PR showed a significantly positive influence on the viability at 5.4 µg/mL. This concentration could be used as the starting point for future tests in vivo. It is worth noting that other authors have reported a reduction of MTT by flavonoids in the absence of viable cells [42,43]. During cell testing we have considered this effect, but could not confirm a significant influence of RU or PR on the absorbance of MTT/formazan at a wavelength of 570 nm in the absence of cells under the conditions investigated.

## 4. Conclusions

The aim of this work was a systematic study of the products obtained by enzymatic polymerization of RU in water, without the addition of organic solvents. Synthesis without organic solvents is important from the point of view of the synthesis of a pure wound healing substance without any toxic residues in the product, which will be in contact with the wound. At the same time, the described method for the production of PR is cheap and it considers the principles of green chemistry—synthesis in water at room temperature, all the raw materials are easily available, the synthesis is catalyzed by enzyme laccase, which does not need any toxic additives or it does not produce any unwanted by-products. The enzyme could also be recovered and reused. 

A comprehensive study of the chemical structure of PR synthesized in water, and its comparison with the chemical structure of PR synthesized in aqueous organic medium from the literature, has shown, that the reaction media has an impact on the chemical structure. Nevertheless, it would be worthwhile to consider the optimization of the process in order to increase the yield of the reaction. This could be possible also by the addition of a natural organic solvent to the reaction mixture (e.g., limonene), which could have an impact on the chemical structure, but, however, it could increase the solubility of RU, and it is also likely to enhance the catalytic performance of laccase [44], and with this it could increase the yield of reaction. At the same time, any residue of such a solvent in the final product could have an additional positive effect on wound healing. 

The investigation of antioxidant and free radical scavenging potential of PR with respect to its chemical structure, has shown, that water-based enzymatic polymerization of RU enhanced or even improved the scavenging activity of biologically important radicals. In addition, water soluble PR had a positive influence on the viability of the human skin derived fibroblasts and HaCaT cells, proving its potential for further studies related to its topological wound healing application, i.e., in the form of bioactive wound dressings. The knowledge on the chemical structure of PR and on its chemical structure–antioxidant activity relationship, obtained in this study, could be in the future transferred also in the new approach of enzymatic phenol polymerization, which has been developed recently and use the genetic manipulation to achieve the displaying of initiating the enzyme, synthesized by the cells themselves, on the cell surface, where the enzyme serves for radical polymerization of added polysaccharides with phenol moieties and encapsulation of the cell themselves [45,46]. The synthesis of PR according to recently described approach could be potentially useful for in vivo cellular therapies and it could represent a new approach for protection of cells against oxidative stress in wound healing therapies.

## Figures and Tables

**Figure 1 polymers-11-01566-f001:**
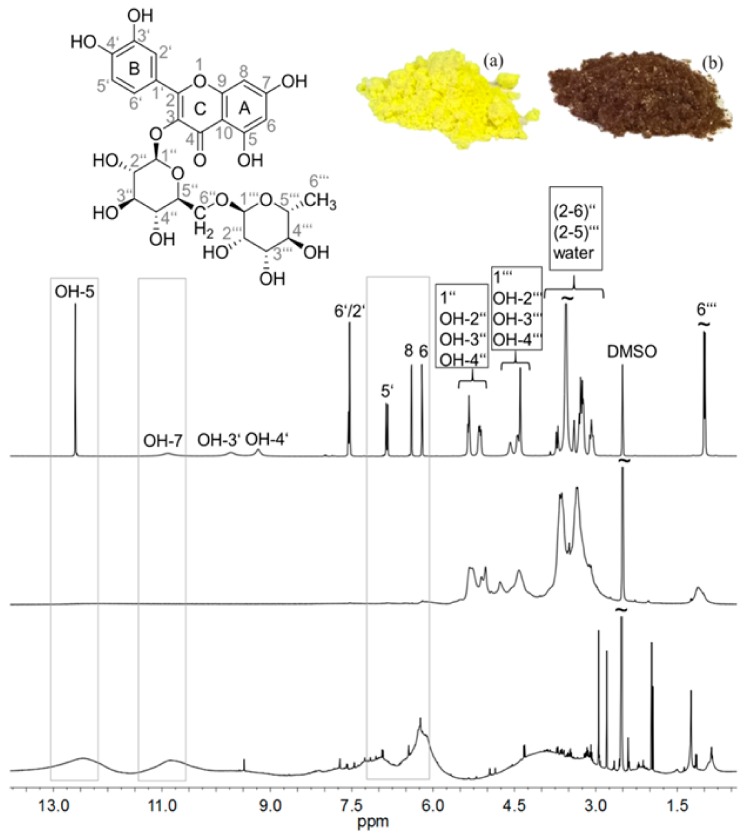
^1^H NMR spectra of rutin (RU; **top**), PR (**centre**), and polyquercetin (PQ; **bottom**) recorded at 400 Mhz in DMSO-d_6_ and digital image of RU (**a**) and PR (**b**).

**Figure 2 polymers-11-01566-f002:**
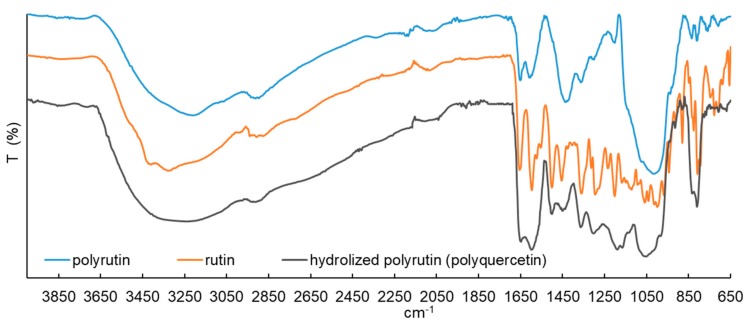
ATR-FTIR spectra of PR (**top**), RU (**centre**), and PQ (**bottom**).

**Figure 3 polymers-11-01566-f003:**
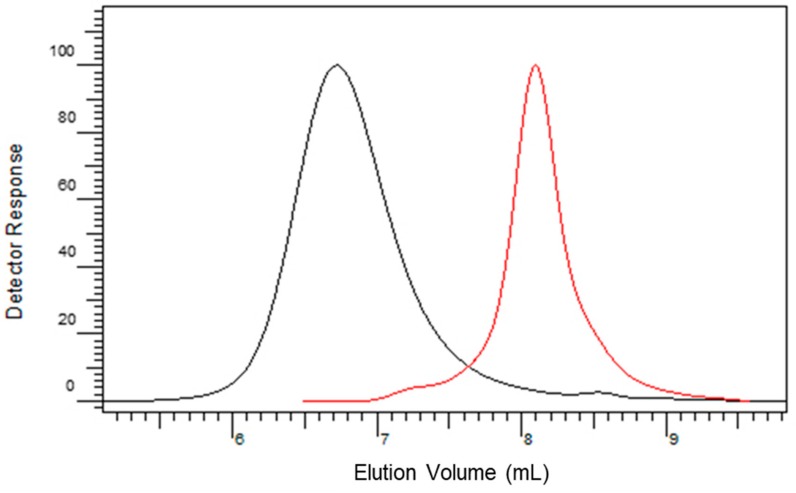
Size exclusion chromatogram of RU (**red curve, right**) and PR (**black curve, left**).

**Figure 4 polymers-11-01566-f004:**
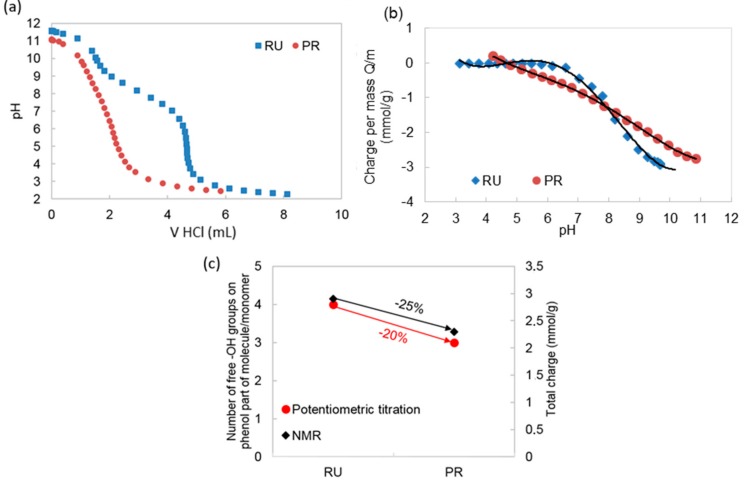
(**a**) Potentiometric charge titration of RU and PR as a function of pH; (**b**) potentiometric charge titration of RU and PR shown as charge per mass as a function of pH; and (**c**) comparison of the number of free phenol hydroxyl groups/monomer, determined using ^1^H NMR, with the total charge determined using potentiometric titration for RU and PR.

**Figure 5 polymers-11-01566-f005:**
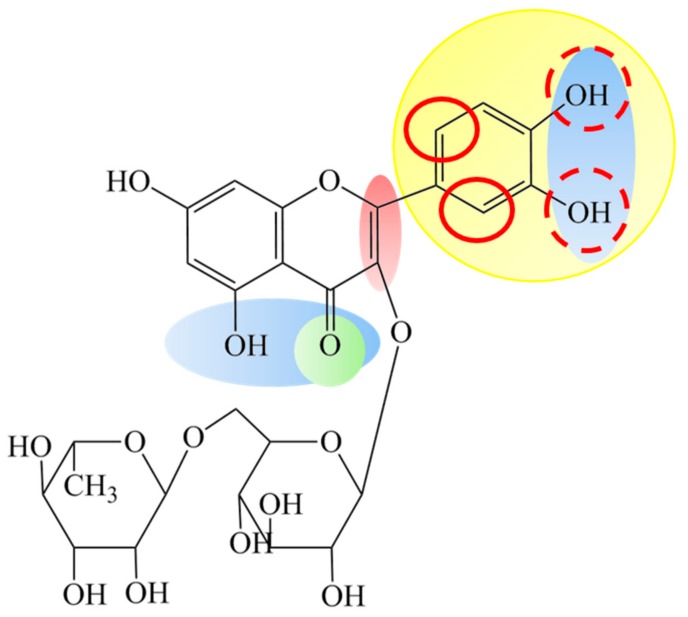
Structure of RU showing the features important in classical antioxidant potential.

**Figure 6 polymers-11-01566-f006:**
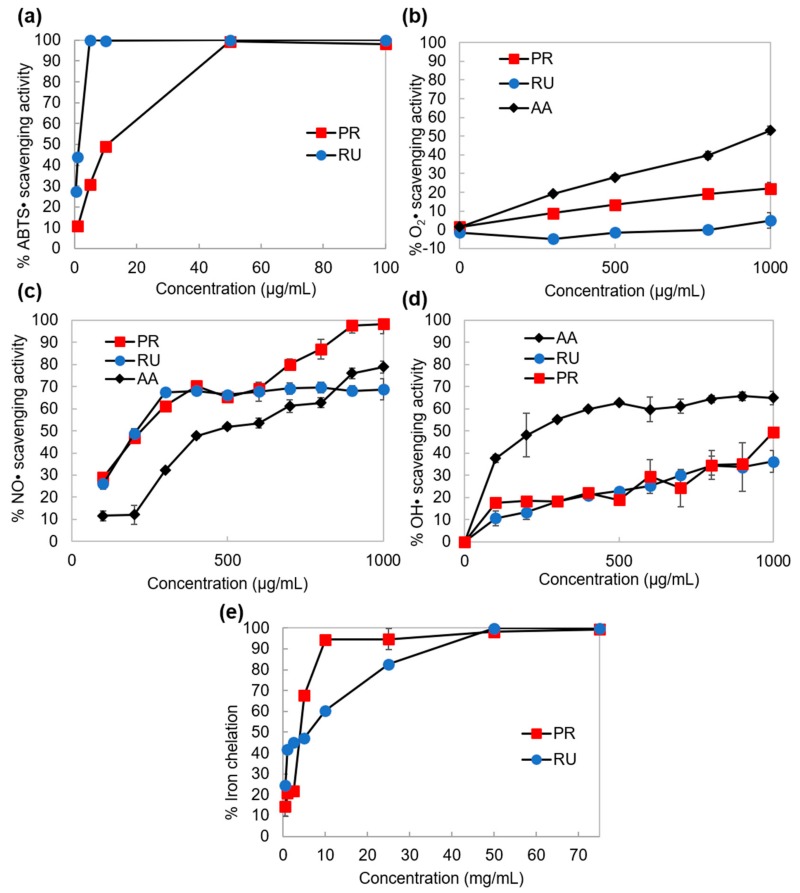
Antioxidant activity of RU and PR determined by the scavenging activity of (**a**) ABTS•, (**b**) O_2_•, (**c**) NO•, (**d**) OH•, and (**e**) by the Fe^2+^-chelating ability of RU and PR. Each value is expressed as a mean ±SD (*n* = 2).

**Figure 7 polymers-11-01566-f007:**
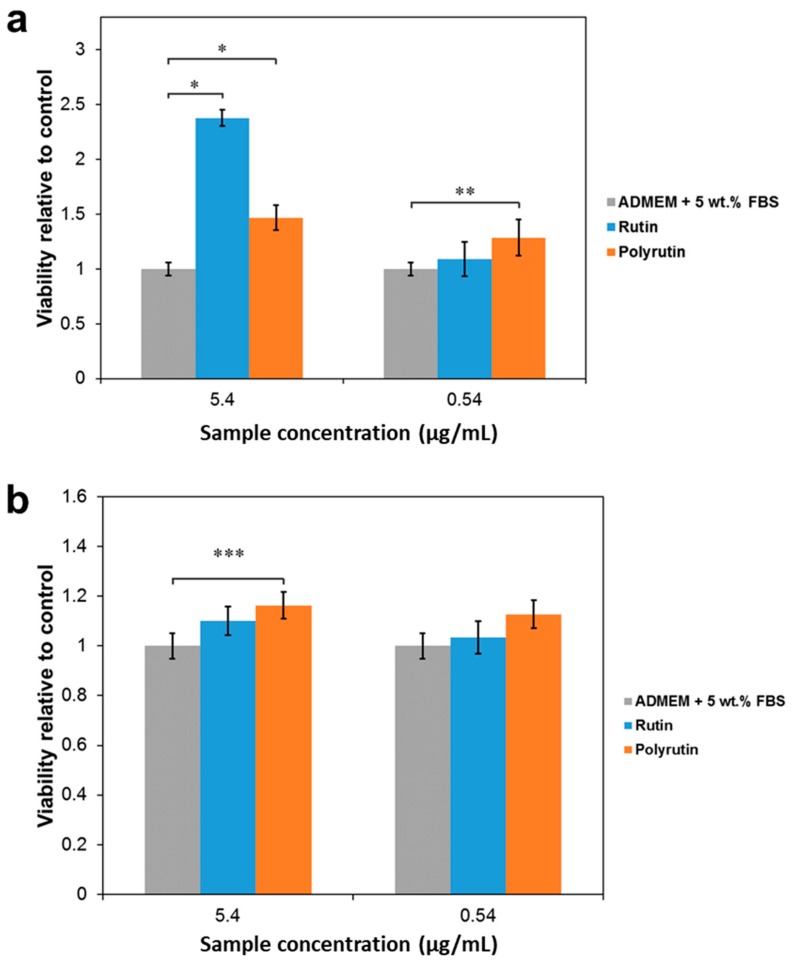
Cell viability obtained from an MTT assay of (**a**) human skin derived fibroblasts and (**b**) HaCaT cells. RU and PR in the cell culture solution at concentrations of 5.4 and 0.54 µg/mL. Values are expressed as percentage of the means ± SD (*n* = 4). Statistical significance was defined as * *p* < 0.001, ** *p* < 0.005, *** *p* < 0.05 compared to the control sample (ANOVA).

**Table 1 polymers-11-01566-t001:** Solubility of polyrutin (PR) synthesized in water media and PR synthesized in aqueous organic media [8] in solvents with a different polarity index [27].

Solvent		Water	DMSO	DMA	DMF	Pyridine	Methanol	Acetone	THF	Toluene
Polarity index		10.2	7.2	6.5	6.4	5.3	5.1	5.1	4.0	2.4
**Solubility Determined Photometrically (mg/mL)**										
Reaction media	Aqueous	753 ± 27	465 ± 15	92 ± 14	104 ± 24	2.4 ± 1	insoluble	insoluble	insoluble	insoluble
**Solubility Determined by Visual Observation by Kurisawa et al. [8]**										
Reaction media	Aqueous organic	**++**	**++**		**++**	**+/−**	**+**	**−**		**−**

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
