# Peer review of "Chemical Structure–Antioxidant Activity Relationship of Water–Based Enzymatic Polymerized Rutin and Its Wound Healing Potential"

_polymers, 2019, doi:10.3390/polym11101566_

Round 1

Reviewer 1 Report

Pivec et. al. report a study on enzymatic polymerization of the flavonoid rutin into poly(rutin) (PR) using laccase. A search of the literature reveals that enzymatic polymerization of rutin into polyrutin has been previously reported, and the authors allude to prior studies using laccase to polymerize PR  (lines 317-318). The experiments appear to have been carefully performed. They carry out simple water-based polymerization of rutin with laccase and perform physico-chemical characterization to try and understand the molecular structure of the resulting PR. They use NMR, FTIR, SEC and titration. SEC showed relatively small polymers of ~15000 Mw and PDI of 2.2.  In the next part of the study they test the antioxidant potential of PR. Finally they report cell viability studies via MTT assay. The work does not report any fundamental new concept, but rather attempts to elucidate the connectivity of PR in the polymers following polymerization with laccase. In my opinion it can be suitable for publication in this journal after the following comments have been addressed.

1.     References: Recently a new concept in enzyme-mediated phenol polymerization was reported. I believe these two papers (given below) are relevant papers that can be added to the references of the current manuscript. They detail a new approach to enzymatic phenol polymerization that is achieved using enzymes synthesized by the cells themselves and displayed on the outer cell wall.

https://doi.org/10.1002/bit.27002

https://doi.org/10.1021/acs.chemmater.8b04348

2.     The physico-chemical characterization is an important part of this study. I was curious to find that the authors did not use mass spectrometry to look at, for example, polydispersity, charge and molecular weight of the samples. If the authors have access to a mass spectrometer (for example, high-res ion spray or maldi) then they should also perform mass spec analysis on the PR. This will add a new level of analysis and increase the quality of the paper without much additional effort.

3.     Panel 6b – the value at which the x-axis crosses the y-axis should be changed to -10 to correspond to the bottom of the plot. This way the number labels will not overlap with the bottom line of the panel.

Reviewer 2 Report

1. why did you select the 5.4 and 0.54 µg/mL to test Cell viability? how do the effect of other sample concentration on Cell viability?

Reviewer 3 Report

General comments

The article deals with the enzymatic polimerization of the important flavonoid rutin in aqueous phase, without organic solvents (complying with green chemistry principles), aimed at otaining higher water solubility of the polymerized rutin, while retaining the bioactive properties relevant to physiological applications, such as wound healing. The use of the enzyme laccase extracted from a non-edible mushroom and the respctive precipitation method are also noteworthy.

The research aims and scope comply with the journal's standards, the experimental design is correct and comprehensive, the results are interesting and - overall - well presented. The study is well documented, is original and adds to the existing knowledge.

In my opinion, few minor changes are needed in order to make the manuscript acceptable for publication; however, the Conclusions section should be substantially rewritten because, in the current form, it is a repetition of information provided elsewhere in the main text, as specified in detail below.

Specific comments

Line 36. "RU". Define abbrevoation at first appearance in the main text too (not only in the Abstract).

Line 55. "the avoidance of organic solvents is, however, desirable". What about the possible use of natural organic solvents, such as limonene, which has strong bioactivity too? (if rutin is enough soluble in limonene...). A very short discussion would add to completeness.

Line 72. Start new paragraph after the subsection title.

Lines 204-206. The statement "After the centrifugation... in this work" is speculative. This study is already enough complex and interesting, thus I suggest removing unsupported statements.

Line 208. The paragraph starting this line concerns the solubility topic, and should be labeled accordingly (such as the label "1H NMR spectroscopy" in the following page).

Line 230. "Table 1". Symbols such as "+", "++" and "-" concerning the solubility are qualitative. Quantitative thresholds should be defined for the sake of clarity.

Lines 360-361. "one phenol hydroxyl group, which is equal to 25 % of phenol hydroxyl group". Unclear, to be rephrased.

Lins 475-476. "The positive effect of RU and PR on the viability of fibroblasts compared to the control was proven at 5.4 µg/mL". However, it should be highlighted that RU is much more effective at this concentration. Any explanation for this?

Line 483. "5. Conclusions". The Conclusions should not merely repeat the discussion of the results, let alone the methods. More succint and attractive conclusions should be drafted, conveying the very important messages resulting from the study, among which the compliance with green chemistry principles, the affordability and reproducibility of the method (short discussion about these topics), the easy acession to the enzyme laccase (or at least to the respective raw material). Repeating all details is useless and somehow confusing. Lines from 509 to 512 are well written, and could serve as a model for more effective concliusions.

Round 2

Reviewer 1 Report

The authors have adqeuately responded to the reviewer'S concerns.